# Retrospective secondary data analysis to identify high-cost users in inpatient department of hospitals in Thailand, a middle-income country with universal healthcare coverage

Waranya Rattanavipapong [iD] ,[1] Yi Wang,[2] Rukmanee Butchon,[1] Nitichen Kittiratchakool,[1] Jadej Thammatacharee,[3] Yot Teerawattananon,[1,2] Wanrudee Isaranuwatchai[1,4]

[1]Health Intervention and Technology Assessment Program, Ministry of Public Health, Nonthaburi, Thailand
[2]Saw Swee Hock School of Public Health, National University of Singapore, Singapore
[3]National Health Security Office, Bangkok, Thailand
[4]Institute or Health Policy, Management and Evaluation, University of Toronto, Toronto, Ontario, Canada

**Correspondence to**
Dr Wanrudee Isaranuwatchai;
wanrudee.i@hitap.net

## ABSTRACT

**Objectives** The study aims to identify high-cost users (HCUs) in the inpatient departments of hospitals in Thailand including their common characteristics, patterns of healthcare utilisation and expenditure compared with low-cost users, and to explore potential factors associated with HCUs so the healthcare system can be prepared to support the HCUs including those who have increased chances of becoming HCUs.

**Design and setting** A retrospective secondary data analysis using hospitalisation data from Thailand's Universal Coverage Scheme (UCS) obtained from the National Health Security Office over a 5-year period from October 2014 to September 2019 (fiscal year 2014–2018).

**Participants** Study participants included Thai citizens who had at least one inpatient admission to hospitals under the UCS over the study period.

**Results** Over the 5-year period, the top 5% of the hospitalised population (or HCUs) consumed almost 50% of the health expenditure each year. HCUs were more likely to have longer hospital stays, a higher annual number of visits and be admitted to multiple hospitals each year when compared with the low-cost users (the bottom 50% of the hospitalised population). The study further reported that the chance of becoming an HCU is associated with several factors such as increasing age, being male, having a comorbidity and being admitted to hospitals in Bangkok.

**Conclusions** This study confirmed that the HCU phenomenon existed in Thailand, where a majority of inpatient care spending is concentrated in the top 5% of the hospitalised population. The study findings call attention to potential initiatives that can help monitor the magnitude and trend of HCUs and develop policies to prevent HCUs.

## INTRODUCTION

There is an increasing emphasis on the sustainability of healthcare systems for many reasons, including the availability of new costly health technologies, the greater demand for healthcare by an ageing

### Strengths and limitations of this study

► This study is one of the first studies to use real-world administrative hospitalisation data from Thailand's Universal Coverage Scheme to explore the high-cost user (HCU) phenomenon, and to identify characteristics of HCUs.

► The study represents an example of how existing routinely collected real-world data can be used to potentially improve care and health service and to help address policy-relevant questions.

► The analysis was limited by data availability, and future work could explore beyond hospitalisation data to consider other types of health services (eg, outpatient services), and could include other potential confounders (eg, education and income).

► Significant work was done to clean and manage the data, which highlighted the importance of data quality especially in countries that may not have used real-world data to their full potential.

population, and the government's commitment to achieve universal health coverage (UHC) under limited resources.[1 2] As such, several solutions have been attempted to enhance the efficiency of healthcare systems around the world.[3–6] These potential solutions include the prioritisation of health problems and associated interventions based on factors such as burden of disease and cost-effectiveness evidence,[7 8] identifying low-value health services that can be omitted without adversely affecting patient outcomes,[9 10] and promoting the use of preventive measures to avoid patients' development of chronic conditions.[11 12]

Based on a literature review of the high-cost user (HCU) phenomenon in several countries, for example, Japan, the USA, Australia

and Canada,[13–18] HCUs are generally defined as the top 5% of patients with the highest health expenditure in a given year, while low-cost users (LCUs) are defined as the bottom 50% of patients with the lowest healthcare costs.[15–17] Although the aforementioned evidence is only from high-income countries, this phenomenon is also relevant and important for low-income and middle-income countries (LMICs) where health resources are limited. Additionally, there are studies on health service utilisation in LMICs focusing on general patterns of utilisation and certain high-cost interventions;[19–21] however they do not focus on HCUs. Therefore, a better understanding of the HCUs in Thailand (an upper-income middle-income country (UMIC))[22] regarding characteristics of HCUs and patterns of healthcare utilisation may help the country (including other LMICs with similar country profiles) to identify potential HCUs and develop more tailored and appropriate interventions to improve the management and financing of these patients.

Thailand has been classified as an UMIC since 2011[22] and has a rapidly growing ageing population. The proportion of the population aged 60 years or over increased from 12% in 2011 to 18% in 2020,[23] and is projected to increase further to 25% in 2030.[24] The leading causes of burden of disease (cause of death and disability) in the Thai population include injury, cardiovascular disease, cancer and diabetes mellitus.[25] These population characteristics, to some extent, align with potential factors associated with HCUs identified in the literature from high-income countries such as being older and having comorbidities.[16 18]

Although the Thai government invests approximately 13.3% of the government budget on public health,[22] the Universal Coverage Scheme (UCS) budget has increased annually during the past decade and at a faster rate than the annual gross domestic product growth, which is directly affecting the sustainability of the healthcare system.[22 26] The three main health insurance programmes employed to cover nearly the entire Thai population include the Civil Servant Medical Benefit Scheme (CSMBS) for government employees and their dependants, the Social Security Scheme (SSS) for formal private sector employees, and the UCS for the remaining Thai citizens.[22] The UCS, which is managed by the National Health Security Office (NHSO), covers approximately 80% of the total population that is not eligible for the CSMBS and SSS, including the low-income or unemployed citizens. Characteristics of the aforementioned three health insurance programmes in Thailand are published elsewhere.[22]

The UCS provides a comprehensive benefits package for curative, rehabilitation, and health promotion and prevention services.[27] Most of the UCS budget is allocated to outpatient and inpatient services. Payment for outpatient care is generally based on capitation while payment for inpatient care is based on diagnosis-related groups (DRGs).[28 29] There is also an additional budget for specific high-cost interventions or medications.[30] Nearly half of the UCS expenditure (eg, 42% in 2011) is spent on inpatient care.[26]

Research on the HCU phenomenon under Thailand's biggest public health insurance scheme, the UCS, is essential for identifying potential measures some of which are deemed 'preventable' spending. For example, spending on treatment for diabetic nephropathy that could have been avoidable if diabetic screening and early treatment had taken place. Moreover, treatment costs for lung cancer could have been avoided if specific public health interventions were available to reduce tobacco consumption.[31 32]

This study aims to identify demographic characteristics of HCUs, understand the common characteristics and patterns of healthcare utilisation and expenditure in hospitals among HCUs as compared with LCUs, and determine potential factors associated with HCUs. This study is the first of its kind in an UMIC and its findings have the potential to be used by healthcare managers and policy makers in relevant settings (including LMICs) to prevent avoidable HCUs in hospitals and reallocate these limited resources for other cost-effective options.

## METHODS
### Study population, setting and data
This study was a retrospective secondary data analysis which examined inpatient department (or hospitalisation) data from Thailand's UCS obtained from the NHSO over a 5-year period from 2014 to 2018. The NHSO defines an inpatient as a patient who was formally admitted into a hospital for treatment and who stayed for a minimum 6 hours. Using this administrative hospitalisation database, the study population included Thai citizens of all ages who were hospitalised under the UCS health insurance scheme and discharged between 1 October 2014 and 30 September 2019, according to the Thai government's fiscal year. A unique national identification (ID) number is registered for each Thai citizen in the system. Thai citizens with a national ID who had at least one hospital admission during the study period were included in the study data set. The subjects' unique IDs were masked using a private key by the data holder before the data set was shared with the research team.

All data were anonymised and de-identified. Records with encrypted identities were checked, and duplicated records or those with incomplete data were excluded from analysis. To ensure that the data were accurate and consistent, conflicting demographic information about date of birth, gender, and admission and discharge dates were investigated and resolved. Conflicting records were removed (with the final number of records being 29 899 719).

Variables considered in the analysis comprised gender, date of birth, primary diagnosis (using ICD-10 codes), admission and discharge dates, discharge status (full recovery, improved, not improved, normal delivery, undelivered, normal child discharged with mother, separation

of mother and baby, stillbirth, and death), types of hospitals (ie, from the lowest level being clinics and community hospitals to the higher level, namely private hospitals, general hospitals, regional hospitals, and other hospitals not under the Ministry of Public Health, such as university hospitals), health regions as shown in the online supplemental figure 1), Charlson Comorbidity Index,[33 34] and reimbursement cost (which refers to the amount estimated to be paid by NHSO and reported in the hospitalisation database).

This study used an existing administrative database (with de-identified individual-level hospitalisation data); therefore, patients had no direct involvement or risk and no consent was required.

## Patient and public involvement
No patient involved.

## Analysis
Using the NHSO's hospitalisation database, HCUs and persistent HCUs were examined using descriptive and multivariable regression analysis. HCUs and LCUs were defined by the percentile of hospitalisation costs accrued. HCUs were individuals who had incurred costs higher than 95% of all users (the top 5%). LCUs were patients in the lowest cost group (the bottom 50%). The data set was treated as repeated cross-sectional when examining HCUs and the analysis was conducted separately by year based on patients' discharge dates using similar methods as seen in the literature.[16 18] For example, if patients were admitted to a hospital in one fiscal year but discharged in the following fiscal year, their costs would be included since the admitted fiscal year. We define persistent HCUs as those who were identified as HCUs for more than 1 year; further explanation is provided below. When examining persistent HCUs, a panel data set was constructed by tracking the individuals' statuses (of being an HCU) over multiple years.

Every hospital admission was tracked within each fiscal year (1 October to 30 September). To address the study aims, first, we sought to determine the concentration of inpatient department expenditures to explore the phenomenon of HCUs in Thailand. Total costs associated with all visits within each fiscal year for each patient (ie, estimated reimbursement costs) were calculated and individuals were sorted into five groups according to the percentile of costs: (1) Above the 99th percentile; (2) Between the 95th and the 98th percentiles; (3) Between the 90th and 94th percentiles; (4) Between the 50th and 89th percentiles; and (5) Below the 50th percentile. Patients may have more than one hospital admission; therefore, we summed up all admissions per patients within each year to calculate the annual hospitalisation cost per patient as used in the HCU grouping.

Next, we examined the differences in demographic characteristics, costs and comorbidities of HCUs as compared with LCUs. Descriptive characteristics of each group were analysed by age, gender and main diagnosis identified based on ICD-10 codes. We also reported healthcare utilisation and expenditure patterns between HCUs and LCUs, specifically number of visits, length of stay (LOS), types of hospitals visited and annual cost per patient.

Multivariable logistic regression was conducted to identify potential factors associated with HCUs. The outcome variable was patient HCU status (yes/no) and the exploratory variables included age, gender, number of visits, types and health regions of hospitals visited, main diagnoses, and comorbidities (measured as Charlson Comorbidity Index, and the presence of the top five diagnoses with highest total cost to the public healthcare payer for the study population namely neoplasms, circulatory diseases, respiratory diseases, digestive problems and injuries). Interaction terms were also included to examine the heterogeneous effects on becoming a potential HCU; specifically, interactions between death, age or gender, and the top five primary diagnoses (neoplasms, circulatory diseases, respiratory diseases, digestive problems and injuries), including between types of hospital and health regions, were considered.

To identify the persistent HCUs, only the data between 2015 and 2017 were examined due to the completeness of the data. To construct a panel data set, patients who were presented in the data set in 2015 were used for the analysis. Any patients who died were excluded as they could not become persistent HCUs. For the remaining patients, we examined their HCU status in the years 2016 and 2017 to determine whether they were persistent HCUs. They were then grouped into three levels: 0 (not an HCU for any of the 3 years), 1 (being an HCU for only 1 year) and 2 (being an HCU for at least 2 years). Group 2 was defined as persistent HCUs. A multinomial logistic regression was applied to examine the potential factors associated with persistent HCUs.

All analyses were conducted using Stata Statistical Software (Release 15, College Station, Texas, USA). Statistical significance was set at a value of $p < 0.05$.

## RESULTS
A total of 29 899 719 admissions (14 464 180 unique patients) between fiscal years 2014 and 2018 were analysed. On average, each year had approximately 5.9 million admissions (about 3.9 million unique patients). Additional details on the data management process can be found in online supplemental figure 2.

The hospitalisation data of UCS have shown that a small proportion of patients (the top 5% of the hospitalised population or HCUs) consumed almost half of the healthcare expenditure; this trend was observed over a 5-year period as presented in figure 1. During this period, patients in the top 1% and 5% cost groups accounted for 16-18% and 45-48% of total spending, respectively. Furthermore, a separate analysis showed that HCUs had higher annual mortality rates compared with LCUs.

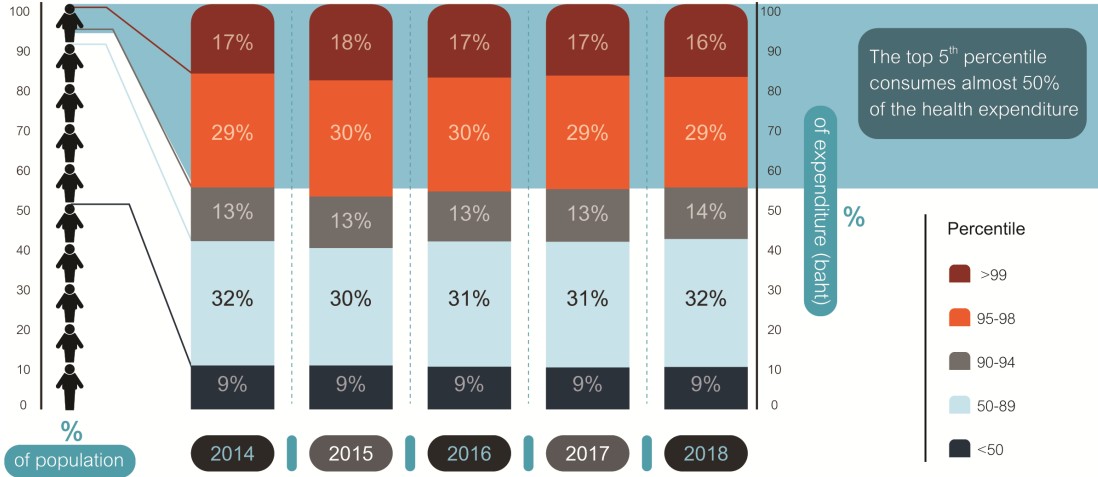

**Figure 1** Distribution of total hospitalisation expenditures across cost percentiles.

Among HCUs, mortality ranged from 19% to 21% over 5 years, compared with 0.1% to 1.3% in LCUs.

To facilitate the understanding of the HCU phenomenon, the profile of HCUs was studied. Generally, the average age of HCUs was 55-56 years, where more than half (50%–54% each year) were in the age group of ≥60 years. In Thailand, about 18% of the population was older adults (≥60 years). Approximately 55% of this sample was male, while approximately 49% was male in the general Thai population.[23] Table 1 provides the descriptive characteristics of HCUs as compared with LCUs.

Over the study period, HCUs were older and included more men compared with LCUs. The mean age ranged between 55-56 years and 23-25 years among HCUs and LCUs, respectively. HCUs had longer LOS and used more healthcare services (higher frequency of hospital visits and higher number of visited hospitals). Moreover, HCUs visited high-level hospitals (eg, regional and university hospitals), while LCUs visited predominantly community hospitals. The patterns of the results were similar across the 5-year period.

Regarding the expenditure pattern, HCUs spent considerably more per year than LCUs in the index year. Over 5 years, the average annual spending for HCUs was between 98 000 (±71 200) baht and 115 000 (±77 400) baht, compared with 2200 (±810) baht and 2900 (±1120) baht for LCUs, where US$1 is approximately 32 Thai baht. Details of the expenditure pattern are presented in table 1.

The most common and most expensive clinical conditions between HCUs and LCUs were examined and found to differ. Table 2 confirms that the expenditures were on different diagnoses. Among HCUs, the most frequent diagnoses included neoplasms and diseases of the circulatory system and respiratory system which consumed more than half of the annual expenditure. Among LCUs, the most frequent diagnoses were broader and included infectious and parasitic diseases, factors influencing health status and contact with health services (eg, admissions for investigation), and diseases of the respiratory system; almost half of the LCUs' annual expenditure was spent on these conditions.

Regression models were used to examine the potential factors associated with being an HCU (table 3). The findings suggest that age, gender, types and zones of hospital admissions, disease and multimorbidity, number of visits, and mortality were associated with becoming an HCU (although the relative contribution of these factors towards the expenditure varied). Most of the interaction terms were statistically significant, suggesting that the impact of these factors was heterogenous among different groups.

On average, HCUs were more likely to be: (1) Older; (2) Male; (3) Admitted to higher-level hospitals (eg, regional and university hospitals); (4) Admitted to hospitals in Bangkok; (5) Patients with a primary malignant tumour, diseases of the circulatory system, injury, poisoning and certain other consequences of external causes; (6) Patients with multiple diagnosis; (7) Patients with other comorbidities; (8) Patients with more than one hospitalisation within a year; and (9) Patients who died before their discharge. The marginal effects are presented in online supplemental table 1, showing that the results, in terms of the probability of being an HCU for average individuals, were in line with the logistic regression. For example, in 2014, an average increase in age by 1 year had a predicted 0.03% higher chance of becoming an HCU. The average patient with neoplasms (a primary malignant tumour) had a predicted 8% higher chance of being an HCU.

The interactions of each of the potential factors associated with HCU status showed that individuals who died before their discharge having had diseases of the respiratory system (OR=1.39–2.08) or of the digestive system (OR=1.04–1.44) were more likely to be HCUs. Moreover, those who were admitted to a general hospital, community hospital or other Ministry of Public Health (MOPH) agencies in Bangkok were also more likely to be HCUs.

This study also examined potential identifying factors of individuals who were likely to remain HCUs over

**Table 1** Demographic characteristics and patterns of utilisation and expenditure of HCUs and LCUs from 2014 to 2018*†

| Characteristic | 2014 | | 2015 | | 2016 | | 2017 | | 2018 | |
|---|---|---|---|---|---|---|---|---|---|---|
| | HCUs | LCUs | HCUs | LCUs | HCUs | LCUs | HCUs | LCUs | HCUs | LCUs |
| Number of patients | 234 214 | 1 913 712 | 235 954 | 1 964 952 | 243 850 | 1 991 451 | 239 180 | 1 953 317 | 243 951 | 1 992 271 |
| Number of admissions | 864 787 | 1 991 875 | 863 308 | 2 054 265 | 899 142 | 2 090 249 | 926 434 | 2 053 590 | 951 527 | 2 096 159 |
| Mean age in years (SD) | 54.6 (22.6) | 22.6 (25.0) | 54.5 (22.4) | 24.1 (25.5) | 55.4 (22.3) | 23.8 (25.4) | 55.6 (22.3) | 24.8 (25.9) | 56.1 (22.1) | 25.2 (26.1) |
| Male (%) | 55% | 44% | 55% | 44% | 55% | 45% | 55% | 45% | 55% | 45% |
| Average length of stay in days (SD) | 11.3 (21.2) | 2.5 (2.8) | 11.3 (22.0) | 2.6 (2.1) | 10.9 (21.6) | 2.6 (2.9) | 10.8 (19.1) | 2.6 (2.6) | 10.4 (18.8) | 2.6 (2.3) |
| Average number of visits (SD) | 3.7 (3.3) | 1.0 (0.2) | 3.7 (3.3) | 1.0 (0.2) | 3.7 (3.4) | 1.0 (0.2) | 3.9 (3.5) | 1.0 (0.2) | 3.9 (3.5) | 1.0 (0.2) |
| Number of hospitals visited (% of admissions) | | | | | | | | | | |
| 1 | 49.4% | 99.0% | 49.7% | 99.0% | 49.8% | 99.0% | 48.5% | 99.0% | 48.4% | 99.0% |
| 2 | 41.3% | 1.0% | 40.4% | 1.0% | 40.7% | 1.0% | 41.6% | 1.0% | 41.8% | 1.0% |
| 3 | 8.1% | – | 8.6% | – | 8.3% | – | 8.7% | – | 8.6% | – |
| >3 | 1.2% | – | 1.3% | – | 1.2% | – | 1.2% | – | 1.2% | – |
| Types of hospitals visited (% of admissions) | | | | | | | | | | |
| Regional hospitals | 35.5% | 16.2% | 33.6% | 16.2% | 34.9% | 15.8% | 34.0% | 16.3% | 34.3% | 16.0% |
| General hospitals | 21.2% | 21.8% | 20.0% | 22.0% | 20.9% | 21.7% | 21.5% | 22.0% | 22.4% | 21.8% |
| Community hospitals | 20.1% | 53.3% | 18.7% | 54.2% | 19.6% | 54.1% | 20.7% | 53.3% | 21.4% | 53.7% |
| Others | 23.2% | 8.7% | 27.7% | 7.6% | 24.6% | 8.4% | 23.8% | 8.4% | 21.9% | 8.5% |
| Average total cost per patient in baht (SD) | 113 500 (75 600) | 2800 (1060) | 98 000 (71 200) | 2200 (810) | 109 200 (76 300) | 2500 (950) | 111 100 (74 500) | 2700 (1040) | 115 000 (77 400) | 2900 (1120) |

*The numbers (except numbers of patients and admissions) presented in the table were rounded off to the nearest whole number.
†All variables are statistically significant (p<0.05).
HCUs, high-cost users; LCUs, low-cost users; SD, standard deviation.

**Table 2** The top five primary diagnoses of HCUs and LCUs over a 5-year study period*

| High-cost users (HCUs) | | | Low cost users (LCUs) | | |
| --- | --- | --- | --- | --- | --- |
| Conditions | % of total admissions | % of total expenditures | Conditions | % of total admissions | % of total expenditures |
| Neoplasms | 22–23 | 17–20 | Certain infectious and parasitic diseases | 17–19 | 15–17 |
| Diseases of the circulatory system | 17–20 | 25–30 | Factors influencing health status and contact with health services | 14–17 | 9–12 |
| Diseases of the respiratory system | 13–14 | 10–12 | Diseases of the respiratory system | 11–14 | 13–17 |
| Diseases of the genitourinary system | 7–8 | 3 | Pregnancy, childbirth, puerperium | 10–12 | 11–14 |
| Diseases of the digestive system | 7 | 5 | Injury, poisoning and certain other consequences of external causes | 9 | 8–9 |

*The findings reported covered the 5-year study period so the lowest percentage and highest percentage during this period were presented.
HCUs, high-cost users; LCUs, low-cost users.

time. In the data, all the persistent HCUs were HCUs for at least 2 years; no patient was found to be an HCU for three consecutive years. From a descriptive analysis, the average annual cost was higher for persistent HCUs (52 867±74 024) as compared with those who were once-off HCUs (24 577±50 495). In a multinomial logistic regression, the conditions that were associated with a higher likelihood of becoming a persistent HCU were similar to the factors associated with HCUs as shown in the online supplemental table 2. However, the magnitude of the impact differed in some cases. For example, the relative risk ratio for neoplasms was 7.9 for HCUs, but was 43.5 for persistent HCUs. This finding suggests that patients with neoplasms were likely to be HCUs, and HCUs with neoplasms were also likely to be persistent HCUs.

## DISCUSSION

There is a need for understanding the characteristics of HCUs with the highest need in terms of consuming healthcare resources and their underlying factors that may be influencing costs, recognising that among all HCUs, some may truly need those health services whereas some may not. This study first confirmed that the HCU phenomenon existed in Thailand, an UMIC, where a majority of inpatient care spending is concentrated in the top 5% of the hospitalised population; therefore, our results are comparable to the existing literature on this phenomenon globally.

The factors associated with HCUs appear to align with those reported in the international literature. Older and male patients were more likely to be HCUs than younger and female patients, as compared with a study by Rosella et al in Canada.[16] HCUs were also likely to have multiple diagnoses, comorbidity conditions and serious types of diagnoses (eg, cancer and cardiovascular diseases), and therefore, they were also likely to be admitted to hospitals

in Bangkok (the capital of Thailand) with more advanced treatments available. Having chronic conditions (eg, cancer and respiratory diseases) also increased the probability of being a persistent HCU (ie, more than 1 year) as found in the published literature of high-income countries (such as countries in Organisation for Economic Co-operation and Development).[17] This study raises an interesting point in setting health priority using disease burden that has relied on morbidity and mortality outcomes in terms of disability-adjusted life years,[35] though this measure did not take into account burden of care that was measured in this study. For example, although injury is one of the highest disease burdens in Thailand,[25] there are different types of injury; and many may incur low cost whereas some have high cost associated with them. Also, injury in Thailand often occurs among the young population and is associated with a lower average number of visits as compared with other diseases. Injury becomes a contributing factor of HCUs when adjusting other explanatory variables as shown in table 3. This is in contrast to cardiovascular diseases and neoplasms, for example, that are high in both disease burden and cost of care.

Furthermore, the actual costs of healthcare resources consumed by these hospitals should be explored to ensure that the expense paid by NHSO using the DRG approach is adequate for developing and maintaining referral systems, and preventing hospitals from rejecting referred patients.[36] Additionally, the DRG creep, which is a method of modifying clinical coding practices to maximise reimbursement, should be investigated as well.[29 37] Knowing the common diagnoses of HCUs could also underline the associated costs or burden of these health conditions. As frequent users of the healthcare system, HCUs would be expected to visit a hospital more than once a year, as the findings have shown. Information on these potential factors associated with HCUs could

**Table 3** Factors associated with being a high-cost user in the Universal Coverage Scheme

| Factors associated with being a high-cost user | 2014 OR (95% CI) | 2015 OR (95% CI) | 2016 OR (95% CI) | 2017 OR (95% CI) | 2018 OR (95% CI) |
|---|---|---|---|---|---|
| Age | 1.030 (1.029 to 1.031) | 1.035 (1.034 to 1.036) | 1.035 (1.034 to 1.035) | 1.030 (1.030 to 1.031) | 1.034 (1.033 to 1.035) |
| Age*Age | 0.99985 (0.99983 to 0.99986) | 0.99980 (0.99979 to 0.99980) | 0.99982 (0.99981 to 0.99983) | 0.99986 (0.99985 to 0.99986) | 0.99984 (0.99983 to 0.99985) |
| Gender (ref: female) | 0.73 (0.71 to 0.74) | 0.77 (0.75 to 0.78) | 0.79 (0.78 to 0.80) | 0.79 (0.78 to 0.80) | 0.82 (0.81 to 0.83) |
| Hospital type (ref: other clinics) | | | | | |
| Regional hospitals | 3.42 (3.37 to 3.47) | 3.45 (3.40 to 3.50) | 3.47 (3.42 to 3.52) | 3.24 (3.19 to 3.29) | 3.16 (3.11 to 3.20) |
| General hospitals, community hospitals and other MOPH agencies | 1.01 (1.00 to 1.03)* | 1.03 (1.02 to 1.05) | 1.00 (0.98 to 1.02)* | 1.02 (1.00 to 1.03) | 1.02 (1.01 to 1.04) |
| Non-MOPH agencies such as university hospitals | 6.60 (6.45 to 6.75) | 9.06 (8.86 to 9.26) | 7.51 (7.35 to 7.69) | 7.21 (7.04 to 7.37) | 6.42 (6.28 to 6.57) |
| Private hospitals | 2.10 (2.03 to 2.18) | 4.06 (3.94 to 4.18) | 3.46 (3.35 to 3.56) | 3.28 (3.17 to 3.40) | 3.15 (3.04 to 3.26) |
| Hospital zone: (ref: hospitals outside Bangkok) | | | | | |
| Bangkok | 3.12 (2.97 to 3.29) | 4.73 (4.51 to 4.97) | 3.80 (3.61 to 3.99) | 3.92 (3.73 to 4.12) | 3.25 (3.09 to 3.42) |
| Primary diagnosis | | | | | |
| Neoplasms | 9.30 (8.78 to 9.85) | 8.96 (8.46 to 9.49) | 9.09 (8.59 to 9.62) | 8.77 (8.28 to 9.28) | 10.05 (9.49 to 10.64) |
| Diseases of the circulatory system | 9.48 (9.01 to 9.97) | 9.60 (9.13 to 10.0) | 10.60 (10.09 to 11.12) | 11.01 (10.49 to 11.55) | 10.96 (10.45 to 11.50) |
| Diseases of the respiratory system | 1.01 (0.98 to 1.05) | 1.00 (0.96 to 1.03) | 1.04 (1.00 to 1.07) | 0.99 (0.96 to 1.02)* | 1.11 (1.07 to 1.15) |
| Diseases of the digestive system | 0.62 (0.59 to 0.65) | 0.61 (0.58 to 0.65) | 0.62 (0.59 to 0.65) | 0.62 (0.59 to 0.65) | 0.61 (0.58 to 0.64) |
| Injury or poisoning and certain other consequences of external causes | 2.41 (2.33 to 2.50) | 2.45 (2.37 to 2.54) | 2.37 (2.29 to 2.46) | 2.10 (2.02 to 2.18) | 2.26 (2.17 to 2.34) |
| Comorbidity (Charlson Comorbidity Index) | 1.071 (1.065 to 1.077) | 1.045 (1.040 to 1.051) | 1.047 (1.041 to 1.053) | 1.035 (1.029 to 1.040) | 1.019 (1.013 to 1.025) |
| Death | 6.53 (6.34 to 6.73) | 5.42 (5.26 to 5.58) | 5.43 (5.28 to 5.59) | 4.73 (4.59 to 4.88) | 4.06 (3.94 to 4.19) |
| Number of primary diagnoses | 1.20 (1.19 to 1.21) | 1.25 (1.24 to 1.26) | 1.24 (1.23 to 1.25) | 1.27 (1.26 to 1.28) | 1.28 (1.27 to 1.29) |
| Number of visits in a year | 1.65 (1.64 to 1.66) | 1.57 (1.56 to 1.58) | 1.55 (1.54 to 1.56) | 1.56 (1.56 to 1.57) | 1.54 (1.54 to 1.55) |
| Neoplasms*Death | 0.31 (0.30 to 0.33) | 0.38 (0.36 to 0.39) | 0.39 (0.37 to 0.41) | 0.39 (0.38 to 0.41) | 0.43 (0.41 to 0.45) |
| Diseases of the circulatory system*Death | 0.36 (0.35 to 0.37) | 0.40 (0.39 to 0.42) | 0.35 (0.33 to 0.36) | 0.39 (0.38 to 0.41) | 0.44 (0.42 to 0.46) |
| Diseases of the respiratory system*Death | 2.00 (1.92 to 2.08) | 1.59 (1.53 to 1.66) | 1.75 (1.68 to 1.81) | 1.50 (1.45 to 1.56) | 1.44 (1.39 to 1.50) |
| Diseases of the digestive system*Death | 1.09 (1.04 to 1.14) | 1.20 (1.15 to 1.26) | 1.12 (1.07 to 1.18) | 1.22 (1.16 to 1.28) | 1.38 (1.31 to 1.44) |
| Injury or poisoning and certain other consequences of external causes*Death | 0.75 (0.71 to 0.79) | 0.64 (0.6 to 0.67) | 0.63 (0.59 to 0.66) | 0.71 (0.67 to 0.75) | 0.76 (0.72 to 0.80) |
| Neoplasms*Age | 0.987 (0.986 to 0.988) | 0.988 (0.987 to 0.988) | 0.986 (0.985 to 0.987) | 0.987 (0.987 to 0.988) | 0.985 (0.984 to 0.986) |
| Diseases of the circulatory system*Age | 0.981 (0.981 to 0.982) | 0.983 (0.983 to 0.984) | 0.983 (0.982 to 0.983) | 0.983 (0.982 to 0.984) | 0.983 (0.982 to 0.984) |
| Diseases of the respiratory system*Age | 1.007 (1.006 to 1.007) | 1.004 (1.004 to 1.005) | 1.005 (1.004 to 1.005) | 1.003 (1.003 to 1.004) | 1.002 (1.001 to 1.00) |
| Diseases of the digestive system*Age | 1.006 (1.006 to 1.007) | 1.005 (1.004 to 1.006) | 1.006 (1.005 to 1.007) | 1.005 (1.004 to 1.005) | 1.004 (1.004 to 1.015) |
| Injury or poisoning and certain other consequences of external causes*Age | 0.999 (0.999 to 1.001)* | 1.001 (1.001 to 1.002) | 1.001 (1.000 to 1.002) | 1.001 (1.001 to 1.002) | 1.000 (0.999 to 1.001)* |
| Neoplasms*Gender | 0.84 (0.82 to 0.87) | 0.75 (0.72 to 0.77) | 0.71 (0.69 to 0.74) | 0.71 (0.69 to 0.73) | 0.66 (0.64 to 0.68) |
| Diseases of the circulatory system*Gender | 1.00 (0.98 to 1.03)* | 0.90 (0.88 to 0.92) | 0.91 (0.89 to 0.93) | 0.90 (0.88 to 0.92) | 0.88 (0.86 to 0.91) |

Continued

**Table 3** Continued

| Factors associated with being a high-cost user | 2014 OR (95% CI) | 2015 OR (95% CI) | 2016 OR (95% CI) | 2017 OR (95% CI) | 2018 OR (95% CI) |
|---|---|---|---|---|---|
| Diseases of the respiratory system*Gender | 0.98 (0.95 to 1.01)* | 0.98 (0.95 to 1.01)* | 0.94 (0.91 to 0.96) | 0.93 (0.91 to 0.96) | 0.91 (0.89 to 0.93) |
| Diseases of the digestive system*Gender | 1.07 (1.04 to 1.11) | 1.06 (1.03 to 1.1) | 1.02 (0.99 to 1.05)* | 1.03 (1.00 to 1.07) | 1.01 (0.98 to 1.05)* |
| Injury or poisoning and certain other consequences of external causes*Gender | 1.17 (1.13 to 1.21) | 1.20 (1.16 to 1.24) | 1.21 (1.18 to 1.25) | 1.23 (1.19 to 1.26) | 1.22 (1.19 to 1.26) |
| Hospital zone: Bangkok*Regional hospitals | 0.62 (0.59 to 0.66) | 0.57 (0.54 to 0.61) | 0.59 (0.56 to 0.63) | 0.59 (0.56 to 0.63) | 0.66 (0.62 to 0.69) |
| Hospital zone: Bangkok*General hospitals, community hospitals and other MOPH agencies | 1.79 (1.71 to 1.88) | 1.88 (1.79 to 1.96) | 1.83 (1.75 to 1.92) | 1.69 (1.61 to 1.76) | 1.70 (1.62 to 1.78) |
| Hospital zone: Bangkok*Non-MOPH agencies | 0.30 (0.29 to 0.32) | 0.23 (0.22 to 0.24) | 0.26 (0.24 to 0.27) | 0.25 (0.24 to 0.27) | 0.30 (0.29 to 0.32) |
| Hospital zone: Bangkok*Private hospitals | 0.54 (0.51 to 0.58) | 0.28 (0.27 to 0.30) | 0.34 (0.32 to 0.36) | 0.33 (0.31 to 0.35) | 0.32 (0.30 to 0.34) |
| Constant | 0.0019 (0.0018 to 0.0020) | 0.0017 (0.0016 to 0.0017) | 0.0017 (0.0017 to 0.0018) | 0.0018 (0.0018 to 0.0019) | 0.0017 (0.0016 to 0.0017) |

*P value not statistically significant (p>0.05).
95% CI, 95% confidence interval; OR, Odds ratio.

assist healthcare professionals and hospitals in creating more tailored treatments and management plans for patients who are likely to become HCUs.

This study includes multiple strengths and limitations for future research. To our knowledge, this study is the first to confirm the HCU phenomenon in UMICs, and it is hoped that the results could be applicable to other settings (with similar country profiles), especially low-income countries which may be facing limited resources and need evidence to improve the efficiency of their healthcare system. Knowing the characteristics of those in need would equip the healthcare system to respond accordingly; for example, healthcare providers could design more tailored and appropriate care to the patients. This research represents an example of how existing administrative databases could be used to answer policy-relevant questions.

This study highlights the importance of data quality in administrative databases. There were issues regarding the completeness of data between 2014 and 2018. Year 2014 was the first year that data were computerised; therefore, the data were not complete as data entry had just started. Claims data from hospitals are usually collected by hospitals through medical charts (which can take time) and entered into NHSO systems at a later date (after being checked for accuracy through the NHSO auditing steps); therefore, the data in 2018 are not completely integrated. Regardless, it is useful to include data from 2014 and 2018 to assist in investigating the magnitude and trend of HCUs. Moreover, the analyses encountered several inaccuracies, such as conflicting sexes or dates of birth of individuals with multiple hospitalisations. Therefore, it is integral that when linking across databases (hospitalisation and death databases), a data quality check is put in place to ensure the accurate merging of data; this step should be standard practice for similar analyses.

Additionally, as a secondary data analysis, this study was limited by data availability. Consequently, there could be additional potential confounders which were not included due to lack of data (eg, education and income); this is a limitation that future research should consider. For example, if we were able to link the database to an outpatient visit database, the analysis may be able to examine potential factors associated with HCUs at the ambulatory-care stage as well, considering rehabilitation and prevention and promotion (P&P) programmes. Consequently, patients who died before their discharge were also determined to have a higher chance of being HCUs. This factor (died before discharged) may not be as informative in assisting the development of treatment plans as it is related to the outcome. In the model without this death factor, other factors (mentioned above) remain significant in helping to identify potential HCUs. Only in-hospital mortality was included as hospital admission data for this study; thus, future research should consider national mortality data as well. In addition, selected high-cost specialty medicines and interventions (such as high-cost medicines E2 access program[30] and stents[38]) were

not included in the analysis. These interventions are procured centrally by NHSO and distributed directly to hospitals. Therefore, they are not included in the claims data and could be explored in future research for more detailed results. Furthermore, the categorisation of diagnoses was achieved based on the DRG system which thus led to rather broad categories. This approach was meant to represent initial exploration on common diagnoses among HCUs; therefore, future research could explore diagnoses in a more detailed manner. The current analysis could not distinguish between those who are in true need (for high-cost treatments) versus those whom the healthcare system can support to avoid unnecessary hospitalisation. Some diseases may incur substantial costs over a lifetime (eg, cystic fibrosis), whereas others may incur very high costs in a short amount of time. Further analysis to incorporate these details could add to the understanding of HCUs and, subsequently, further assist the planning of healthcare resources, including highlighting the importance of having a data system with long-term data which allows for exploration of long-term health and economic impacts.

These findings could be applied to several scenarios to support policy-making processes. For example, knowing who the HCUs are could help health professionals monitor the magnitude and trend of HCUs and consider the HCU phenomenon when developing relevant policies such as UCS reimbursement policies to reduce avoidable HCUs or improve the outcomes of high-cost interventions. Moreover, a mechanism could be built to support tertiary hospitals in identifying HCUs on admission and developing measures to support them on a routine basis as opposed to ad hoc monitoring. This mechanism can also make it easier to consider whether to provide aggressive treatment to patients with low probabilities of survival (eg, higher Charlson Comorbidity Score or with multiple organ failure), especially in palliative care management. Furthermore, public communications should be made to individuals in society to raise awareness and encourage the prevention of risk factors through publicly funded P&P programmes in support of avoiding future comorbidities or other chronic diseases (eg, cancer).[39]

In addition, in the healthcare system without UHC, the Thai government could explore the possibility having people with risk factors of HCUs to be one of the priority groups so they can monitor and ensure that the patients are receiving appropriate care before they become HCUs. Low-value care is also of concern in healthcare systems,[9 10 40] and a study of its usage among HCUs could explore the significance of these low-value care services in this population. Further work on referral systems across different levels of health facilities and other types of services (in addition to hospitalisation), including how to provide appropriate care at the right time and right place, could provide evidence to assist policy makers in their planning and delivery of health services. Exploration of HCUs and their transitions across years could further develop the understanding of those with expensive necessities.

## CONCLUSION

This study confirmed the HCU phenomenon in inpatient department of hospitals in Thailand under UCS and reported the characteristics of HCUs. Factors contributing to being HCUs were being male, older, having comorbidities, longer hospital stays, higher annual number of visits, being admitted to multiple hospitals, and having primary diagnoses such as neoplasms, diseases of the circulatory system, injury, or poisoning. Currently, HCUs (the top 5% of hospitalised patients) consumed almost 50% of the healthcare expenditure each year in Thailand. With a better understanding of HCUs, healthcare managers and policymakers may be more equipped to assist patients by providing more tailored and targeted interventions.

**Acknowledgements** The authors thank the National Health Security Office (including Ms Kanjana Sirigomon, Ms Jutatip Thungthong and Mr Poonchana Wareechi) for providing the data and support for this study; Dr Lou Jing, a research fellow at Health Intervention and Policy Evaluation Research, Saw Swee Hock School of Public Health, National University of Singapore, for providing support for the data cleaning steps; and Mr Sarin KC from the Health Intervention and Technology Assessment Program for proofreading support.

**Contributors** The study was conceptualised by YT and WI. WR, YW, RB, NK and JT contributed to its design. Funding was acquired by YT. JT was instrumental in providing data access. The analyses and interpretation were carried out by WR, YW, RB, under supervision of YT and WI. NK provided assistance with presentation of the findings. The initial draft was written by WR and WI, and refined with input from YW, RB, NK, JT and YT. All authors read and approved the final manuscript.

**Funding** This study was supported by the Thailand Research Fund under the senior research scholar on Health Technology Assessment (grant number RTA5980011) and the Bill & Melinda Gates Foundation, the UK Department for International Development, and the Rockefeller Foundation (grant number OPP1202541) through the International Decision Support Initiative (https://www.idsihealth.org/).

**Disclaimer** The findings, interpretations and conclusions expressed in this article are of the authors and do not necessarily reflect the views of the aforementioned funding agencies.

**Competing interests** None declared.

**Patient consent for publication** Not required.

**Ethics approval** The study did not involve any human participants and a research ethics approval was obtained for the secondary data analysis by the Institute for the Development of Human Research Protections, Ministry of Public Health, Thailand (IHRP No.795/2561 on 22 November 2018). The research complied with the non-disclosure agreement between the National Health Security Office and the Health Intervention and Technology Assessment Program to protect individual identification.

**Provenance and peer review** Not commissioned; externally peer reviewed.

**Data availability statement** Data were obtained from a third party and are not publicly available. The data are not publicly available due to privacy or ethical restrictions according to the data sharing agreement with the National Health Security Office.

and indication of whether changes were made. See: https://creativecommons.org/licenses/by/4.0/.

**ORCID iD**

Waranya Rattanavipapong http://orcid.org/0000-0001-8628-2149

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
