## [Reviewer comments · BMJ Open]

ARTICLE DETAILS

TITLE (PROVISIONAL)	Retrospective secondary data analysis to identify high-cost users in inpatient department of hospitals in Thailand, a middle-income country with universal healthcare coverage
AUTHORS	Rattanavipapong, Waranya; Yi, Wang; Butchon, Rukmanee; Kittiratchakool, Nitichen; Thammatacharee, Jadej; Teerawattananon, Yot; Isaranuwachai, Wanrudee

VERSION 1 – REVIEW

REVIEWER	Andrew Argent Red Cross War Memorial Childrens Hospital, Paediatric Intensive Care Unit
REVIEW RETURNED	18-Feb-2021

GENERAL COMMENTS	General Comments Thailand is currently a Higher middle income country, and so it may be important throughout the document to align this data with that group of countries rather than the LMICs which the authors have done. It would be very useful to discuss the concepts of high-cost users in the context of: the age profile of the population (both currently and projected over time). If (as shown in the article) the costs are age related, then it would make a lot of sense to focus on interventions that could be directed at health-improvement in younger age-groups. In that context it would have been very useful to have some insight into the rates of conditions such a Diabetes Mellitus and hypertension. The document will require editing to deal with all the lack of spacing between words, but that is primarily a formatting issue. Specific Comments Title I would suggest that the authors rethink the title of the article. The term “search for an expensive necessity” does not really make sense. Introduction The introduction reads well and provides a good basis for the study. At some point in the paper it may be useful to highlight Thailand’s status as a higher middle income country, with a rapidly aging population. The increasing age of the population may be an important factor in the planning of healthcare services. Historically features such as age, hypertension, nutritional status and obesity, have been associated with high morbidity. It would be
--

really interesting to have a little more insight into the impact of long-term healthcare conditions such as endocrine problems

Methods

Study Population, setting and data

The authors state that: "All data were anonymized and de-identified. Records that were duplicated or had incomplete data were excluded from analysis." It would be very useful to understand how patients are identified in the system. As an example, do all patients have a national identity number that is linked to their hospital admission? Presumably one has to first identify any duplicates, and then move on to de-identification of records.

Would it be possible to link the different categories of patient-care costs to the population distribution? If for instance the costs are very substantial in the elderly when they make up a very small proportion of the population, that has different implications to situations where the elderly make up a very substantial proportion of the population.

It would be very useful to have a sense of the total dataset; the number of duplicates removed; the number of patients with missing data; those entered into the analysis. In the case of missing data, were those patients simply excluded, or was there some effort at interpolation of missing data.

It is a pity that diagnoses have been categorized into such broad groups. As an example circulatory system disorders could include a wide variety of conditions, and the conditions are likely to be very different in younger and older patients.

Analysis

How did the authors deal with patients who were admitted to hospital within one fiscal year, and only discharged in the following fiscal year? How were those costs evaluated?

Were patients transferred from hospital to hospital (e.g. from clinic via regional hospital to tertiary hospital)? If so, was that one visit for that illness, or were there multiple hospital admissions counted?

How were mortality rates calculated? Are these in-hospital mortality, or was there a way of linking hospital admission data to national mortality data?

Results

Would it be possible to analyse the factors associated with high cost healthcare users in more detail? One issue is that many of the factors are associated with each other, and some form of multivariable analysis would probably be more appropriate than reviewing multiple single variables.

Discussion

The authors start the discussion with the statement that: "There is a need for understanding the characteristics of HCUs who are in highest need in terms of consuming health care resources and their underlying factors that may be driving costs." That is an important issue. There are Healthcare users who are consuming very high levels of resources – however do they "need" that expenditure, and are those resources being allocated effectively and reasonably? Is there benefit associated with that expenditure (both for patients and for the society)?

	It may be interesting to discuss the term “high-cost healthcare users” in a slightly broader context. Generally the term relates to individuals who incur substantial healthcare costs over a relatively short period of time. However there is another group of patients (e.g. those with complex long-term healthcare conditions such as cystic fibrosis) who are likely to incur substantial costs over a lifetime related to predominantly outpatient care. From a resource allocation perspective, those patients should be considered. Alternatively there are patients (e.g. neonates with congenital heart disease) who may incur very high costs in the short term (with intensive care, surgical interventions etc), but over a lifetime may incur much lower costs. The approach to these sorts of patients would be very different at a health care level. It would also be interesting to explore the differences between patients who were high-cost users over many years, vs those that have high costs within a particular year, but then have very low costs over a period of time. Unfortunately the data presented do not help to cast light on the issue of how investment into different phases of healthcare (e.g adolescent and young adult care) might have long-term beneficial effects on the overall costs of healthcare. The authors have highlighted some of the challenges related to the data collection their setting, but it would be very helpful if they could provide some thoughts on how the data collection systems could be improved in order to provide high quality data and help to drive decision making. Tables Table 1 The authors have chosen to display the data from each year across the 5 years. Is there a particular reason for that? Is there any trend that is apparent from the period under review? Table 3 I don’t think that the authors are correct in saying “Factors Predicting Being High-cost User”. They are saying (as in the title) that these factors are associated with high cost users. As a specific example death cannot be a predictor of being an high-cost healthcare user. Figures Figure 1 provides some very important information, but I am not sure that it is shown in the best way. I wonder if this could be revisited.
--	---

REVIEWER	Felicia Marie Knaul University of Miami, Institute for Advanced Study of the Americas
REVIEW RETURNED	29-Mar-2021

GENERAL COMMENTS	Review for: “High-cost healthcare users in a middle-income country with universal healthcare coverage: The search for an expensive necessity.” The paper adds to existing knowledge because it is able to characterize High-cost healthcare users (HCUs) in Thailand. The characterization can be used to inform targeted strategies with the purpose of preventing HCUs and utilization disparities under
---

schemes that are committed to universal health coverage (UHC) – especially useful bc Thailand has been an innovative in UHC. The analysis and results presented are also useful for other LMIC because it shows how administrative data can be used to provide an evidence base for decisions around improving utilization and enhancing prevention.

While the BMJ Open is the appropriate place to publish for these reasons, the reviewers' do have serious concerns with the manuscript. The terminology and categorization of HCUs is innovative but the content and theoretical identification of the burden of high-cost users in a health system is not. The reviewers urge the authors to integrate this literature into the contextualization of HCUs. The article does not read well; there are grammatical errors throughout the work and a serious lack of references. There are significant improvements to be made in terms of structure, clarity and contextualization of findings in the discussion section. The discussion section should be re-worked and the conclusions are missing

We suggest major the authors have a chance to resubmit a complete paper with major revisions.

Specific points are further discussed below.

1. Title should specify that the paper is about Thailand
2. Abstract –
 - a. The Interesting and relevant findings are well presented
 - b. Objectives do not read well - convoluted
3. Introduction.
 - a. Structure of the introduction must be improved and streamlined, particularly regarding the structure of the Thai health system.
 - b. Please restate research questions as clear statements
 - c. REFERENCES:
 - i. While the references that are used are appropriate, many references are lacking throughout the section and the entire paper, beginning with the first couple of sentences.
 - ii. We suggest including literature on the disparities in utilization among users within the context of UHC. There are important studies on utilization that need to be referenced as well as integrating literature on high-cost patients. Example: Peltzer, Karl, et al. "Universal health coverage in emerging economies: findings on health care utilization by older adults in China, Ghana, India, Mexico, the Russian Federation, and South Africa." *Global health action* 7.1 (2014): 25314. There is also extensive literature on high-cost patients.
4. Methods

	We suggest including: specific age range, total number of participants in text  a. Is the sample representative? b. Reference Charlson comorbidity index c. Do you use the same methodology as the papers you reference in the intro to categorize and characterize HCUs? Please reference to show systematic methodology. 5. Results  a. Very interesting results. b. We suggest no moving analysis of results “comparable to other studies” to the discussion section. c. State that Table 1 provides descriptive characteristics d. Clarify if differences you are finding (e.g. HCUs are older and LCU are younger) are significant. Suggest adding p-values to Table 1. e. Table 3 shows interesting results and is well presented 6. Discussion - Much of the discussion lacks a clear message. From the reading of the Introduction it sounds like the discussion will focus on contextualizing findings to inform ‘preventable spending’ policies. This connection could be much clearer. Recommendations could also be more punctual and not just a list. We suggest significant re-working of this entire section. 7. Conclusion – There is no conclusion
--	---

VERSION 1 – AUTHOR RESPONSE

Reviewer 1:

Comment: Thailand is currently a upper middle-income country, and so it may be important throughout the document to align this data with that group of countries rather than the LMICs which the authors have done.

Response: *We have revised the “Introduction” section and clarified that Thailand is an upper-middle income country. We tried to explore HCU literature in the low- and middle-income countries (LMICs) context. However, to our knowledge, there is no study about HCUs from LMICs and the study in Thailand will be the first study in LMICs which explored this issue. We hope that our study could be seen as an example and may be applicable to other countries with similar country profiles. Thailand has gained the status of upper-middle income country in 2011. We also added further clarification in the “Discussion” section (3^d paragraph) to clarify this point further.*

Comment: It would be very useful to discuss the concepts of high-cost users in the context of: the age profile of the population (both currently and projected over time). If (as shown in the article) the costs are age related, then it would make a lot of sense to focus on interventions that could be directed at health-improvement in younger age-groups. In that context it would have been very useful to have some insight into the rates of conditions such a Diabetes Mellitus and hypertension.

Response: Thank you for your suggestion. The “Introduction” section (3rd paragraph) was revised to address these points by providing more details about the study setting, Thailand, including potential predictors of HCUs.

Comment: The document will require editing to deal with all the lack of spacing between words, but that is primarily a formatting issue.

Response: The manuscript was edited and proofread carefully by a professional through service suggested by BMJ.

Comment: Title - I would suggest that the authors rethink the title of the article. The term “search for an expensive necessity” does not really make sense.

Response: The title of the manuscript was revised to “Retrospective secondary data analysis to identify high-cost healthcare users in Thailand, a middle-income country with universal healthcare coverage”.

Comment: The introduction reads well and provides a good basis for the study. At some point in the paper it may be useful to highlight Thailand’s status as a higher middle income country, with a rapidly aging population. The increasing age of the population may be an important factor in the planning of healthcare services.

Response: We revised the “Introduction” section (2nd and 3rd paragraphs) to state that Thailand is classified as upper middle-income country.

Comment: Historically features such as age, hypertension, nutritional status and obesity, have been associated with high morbidity. It would be really interesting to have a little more insight into the impact of long-term healthcare conditions such as endocrine problems

Response: We have added more information about the burden of disease in Thai population in the “Introduction” section (3rd paragraph) which linked to common primary diagnoses among HCUs and expenditures on these conditions, and have elaborated more in the “Discussion” section (2nd paragraph).

Comment: The authors state that: “All data were anonymized and de-identified. Records that were duplicated or had incomplete data were excluded from analysis.” It would be very useful to understand how patients are identified in the system. As an example, do all patients have an national identity number that is linked to their hospital admission? Presumably one has to first identify any duplicates, and then move on to de-identification of records.

Response: We have added more details on that the patients were identified, i.e., using their national identification cards, and their ID numbers were anonymized (masked using a private key by the data holder) before the dataset was shared with the research team (in the “Study population, setting, and data” section, 1st paragraph).

Comment: Would it be possible to link the different categories of patient-care costs to the population distribution? If for instance the costs are very substantial in the elderly when they make up a very small proportion of the population, that has different implications to situations where the elderly make up a very substantial proportion of the population.

Response: *The information was added in the “Results” section before Table 1.*

Comment: It would be very useful to have a sense of the total dataset; the number of duplicates removed; the number of patients with missing data; those entered into the analysis. In the case of missing data, were those patients simply excluded, or was there some effort at interpolation of missing data.

Response: *All details are listed in the supplementary material (Table A2) and we mentioned this in the “Results” section, 1st paragraph.*

Comment: It is a pity that diagnoses have been categorized into such broad groups. As an example circulatory system disorders could include a wide variety of conditions, and the conditions are likely to be very different in younger and older patients.

Response: *We thank the reviewer for this good suggestion. We have elaborated on this point in the “Discussion” section, mainly that the categorization of diagnoses was done based on diagnosis-related group (DRG) system, and thus led to rather broad categories. This approach was meant to represent initial exploration on common diagnoses among HCUs. Future research could explore diagnoses in a more detailed manner.*

Comment: How did the authors deal with patients who were admitted to hospital within one fiscal year, and only discharged in the following fiscal year? How were those costs evaluated?

Response: *The analysis was conducted separately by year based on patient’s discharge date. For example, if patients were admitted to a hospital in one fiscal year but discharged in the following fiscal year, their costs would be included since the admitted fiscal year. We elaborated further in the “Analysis” section, 1st paragraph.*

Comment: Were patients transferred from hospital to hospital (e.g. from clinic via regional hospital to tertiary hospital)? If so, was that one visit for that illness, or were there multiple hospital admissions counted?

Response: *Patients could be transferred between hospitals. We summed up all the hospitalization costs for each patient in the same year. Therefore, all hospitalizations from each patient were included. We have clarified this point in the “Analysis” section, 2nd paragraph.*

Comment: How were mortality rates calculated? Are these in hospital mortality, or was there a way of linking hospital admission data to national mortality data?

Response: *The mortality rate was focused only on in-hospital mortality. Good suggestion on the linking of hospital admission data to national mortality data. Unfortunately, that linkage was not possible. We have added this as another limitation in the “Discussion” section.*

Comment: Would it be possible to analyse the factors associated with high cost healthcare users in more detail? One issue is that many of the factors are associated with each other, and some form of multivariable analysis would probably be more appropriate than reviewing multiple single variables.

Response: *We used multivariable regression analysis in this study to analyse potential factors associated with HCUs. More details are provided in the “Analysis” section. As a secondary data analysis, the study was limited by data availability. Consequently, there could be additional potential confounders which could not be included due to lack of data (e.g., education and income), an area where future research could build on. We have explained in the “Discussion” section as a limitation.*

Comment: The authors start the discussion with the statement that: “There is a need for understanding the characteristics of HCUs who are in highest need in terms of consuming health care resources and their underlying factors that may be driving costs.” That is an important issue. There are Healthcare users who are consuming very high levels of resources – however do they “need” that expenditure, and are those resources being allocated effectively and reasonably? Is there benefit associated with that expenditure (both for patients and for the society)?

Response: *We thank the reviewer for this insightful point. We have clarified in the “Discussion” section that we recognize that among all HCUs, some may truly need those health services whereas some may not. Current study aims to first confirmed this HCU phenomenon in this study setting. Future work could further explore, separate, between those who are in true need versus those our health care system can support to avoid unnecessary visits, as further mentioned in the limitations paragraph.*

Comment: It may be interesting to discuss the term “high-cost healthcare users” in a slightly broader context. Generally the term relates to individuals who incur substantial healthcare costs over a relatively short period of time. However there is another group of patients (e.g. those with complex long-term healthcare conditions such as cystic fibrosis) who are likely to incur substantial costs over a lifetime related to predominantly outpatient care. From a resource allocation perspective, those patients should be considered. Alternatively there are patients (e.g. neonates with congenital heart disease) who may incur very high costs in the short term (with intensive care, surgical interventions etc), but over a lifetime may incur much lower costs. The approach to these sorts of patients would be very different at a health care level.

Response: *Another great observation and point to consider. The available data were 5 years which limited our ability to explore the long-term impact/cost. With what we had, we explored the persistent HCUs. We understand the importance of this point and elaborated further in the*

“Discussion” section where further analysis to incorporate these details could add to the understanding of HCUs and subsequently further assist the planning of health care resources including to support the data system which can provide long-term data.

Comment: It would also be interesting to explore the differences between patients who were high-cost users over many years, vs those that have high costs within a particular year, but then have very low costs over a period of time.

Response: *Patients who were HCUs for many years were considered persistent HCUs in this analysis. We explored predictors which could increase the probability of patients becoming persistent HCUs. We have added a part to descriptively explore the average annual health care costs between persistent HCUs and one-time HCUs in the “Results” section. In brief, the persistent HCUs reported higher average cost than one-time HCUs. Please note that this persistent HCUs analysis excluded those who died during the study period as they would not have a chance to become persistent HCUs.*

Comment: Unfortunately the data presented do not help to cast light on the issue of how investment into different phases of healthcare (e.g adolescent and young adult care) might have long-term beneficial effects on the overall costs of healthcare.

Response: *The life cycle approach to explore different phases (e.g. early childhood, adolescent) in term of existing supports from a health care system and gap in health services including their short- and long-term impact on the patients and system would be great topics for future research.*

Comment: The authors have highlighted some of the challenges related to the data collection their setting, but it would be very helpful if they could provide some thoughts on how the data collection systems could be improved in order to provide high quality data and help to drive decision making.

Response: *As one of the first analyses to use this dataset in the study setting, there were many lessons learned about the data for future work. The data were collected by each hospital throughout the country and sent to the National Health Security Office from where we obtained the data. Therefore, the data collection system was already set up and may not be as flexible to modify. With that being said, the data management process including data quality check (as highlighted in the “Discussion” section) could be carried out by the research team before the analysis to ensure the high quality data.*

Comment: Table 1-The authors have chosen to display the data from each year across the 5 years. Is there a particular reason for that? Is there any trend that is apparent from the period under review?

Response: *We would like to share the trend overtime, specifically that a similar trend was found. Currently, this format has been in used previous literature as referenced in the “Introduction” section.*

Comment: Table 3-I don't think that the authors are correct in saying "Factors Predicting Being High-cost User". They are saying (as in the title) that these factors are associated with high cost users. As a specific example death cannot be a predictor of being an high-cost healthcare user.

Response: *We have updated the word predicting to associated as suggested. The predictor (died at discharged) may not be as informative in assisting the development of treatment plan as it is related to the outcome. We ran the analyses for both with and without 'death at discharge' and found the results to be similar. We elaborated on this point in the "Discussion" section, 5th paragraph.*

Comment: Figure 1-provides some very important information, but I am not sure that it is shown in the best way. I wonder if this could be revisited.

Response: *There are a few information we would like to summarize in this Figure 1, i.e., the number of population, the different population percentile, the proportion of hospitalization cost, and the five year timeframe. We have explored a few ways to present several pieces of information in one figure and found that this format is rather informative. Also, this format has been in used previous literature as referenced in the "Introduction" section.*

Reviewer 2:

The paper adds to existing knowledge because it is able to characterize High-cost healthcare users (HCUs) in Thailand. The characterization can be used to inform targeted strategies with the purpose of preventing HCUs and utilization disparities under schemes that are committed to universal health coverage (UHC) – especially useful bc Thailand has been an innovative in UHC. The analysis and results presented are also useful for other LMIC because it shows how administrative data can be used to provide an evidence base for decisions around improving utilization and enhancing prevention. While the BMJ Open is the appropriate place to publish for these reasons, the reviewers' do have serious concerns with the manuscript. The terminology and categorization of HCUs is innovative but the content and theoretical identification of the burden of high-cost users in a health system is not. The reviewers urge the authors to integrate this literature into the contextualization of HCUs. The article does not read well; there are grammatical errors throughout the work and a serious lack of references. There are significant improvements to be made in terms of structure, clarity and contextualization of findings in the discussion section. The discussion section should we re-worked and the conclusions are missing

We suggest major the authors have a chance to resubmit a complete paper with major revisions. Specific points are further discussed below.

Comment: Title should specify that the paper is about Thailand

Response: *The title of the manuscript was revised to "Retrospective secondary data analysis to identify high-cost healthcare users in Thailand, a middle-income country with universal healthcare coverage".*

Comment: Abstract–The Interesting and relevant findings are well presented

Response: *Thank you.*

Comment: Objectives do not read well – convoluted

Response: *We have revised the last paragraph of the “Introduction” section.*

Comment: Structure of the introduction must be improved and streamlined, particularly regarding the structure of the Thai health system.

Response: *The manuscript was edited and proofread carefully by a professional through service suggested by BMJ.*

Comment: Please restate research questions as clear statements

Response: *We have revised the last paragraph of the “Introduction” section.*

Comment: While the references that are used are appropriate, many references are lacking throughout the section and the entire paper, beginning with the first couple of sentences.

Response: *The manuscript was edited and proofread carefully by a professional through service suggested by BMJ.*

Comment: We suggest including literature on the disparities in utilization among users within the context of UHC. There are important studies on utilization that need to be referenced as well as integrating literature on high-cost patients. Example: Peltzer, Karl, et al. "Universal health coverage in emerging economies: findings on health care utilization by older adults in China, Ghana, India, Mexico, the Russian Federation, and South Africa." *Global health action* 7.1 (2014): 25314. There is also extensive literature on high-cost patients.

Response: *We thank the reviewer for suggesting the literature and have added accordingly. There are indeed a great deal of literature on health care utilization including high burden diseases or interventions in LMICs. With that being, the literature specifically on high-cost users in LMICs remain limited. We have elaborated this point further in the “Introduction” section. This is another reason we would like to publish our study with the hope that it could be an example and encourage other country (especially LMICs) to fill in this gap in the literature.*

Comment: We suggest including: specific age range, total number of participants in text. Is the sample representative?

Response: *Thailand has three public health insurance schemes and this study focused on one health insurance schemes which covered approximately 80% of the population. The study sample included all patients who were admitted to a hospital in Thailand under a universal coverage scheme (public health insurance) during the study period. Therefore, the sample should be representative to patients receiving this one type of health insurance from the government. We have added more details on the requested information in the “Introduction” section.*

Comment: Reference Charlson comorbidity index

Response: *We have added references for Charlson comorbidity index.*

Comment: Do you use the same methodology as the papers you reference in the intro to categorize and characterize HCUs? Please reference to show systematic methodology.

Response: *We did and have added the reference in the methods.*

Comment: Very interesting results.

Response: *Thank you.*

Comment: We suggest no moving analysis of results “comparable to other studies” to the discussion section.

Response: *Thank you and we have updated accordingly.*

Comment: State that Table 1 provides descriptive characteristics

Response: *We have updated the text as you suggested.*

Comment: Clarify if differences you are finding (e.g. HCUs are older and LCU are younger) are significant. Suggest adding p-values to Table 1.

Response: *Table 1 provides descriptive characteristics of HCUs and LCUs and we added the p-values as suggested . The factors contributing to HCUs (including age) were modelled and reported in Table 3 and the Supplementary Appendix.*

Comment: Table 3 shows interesting results and is well presented

Response: *Thank you.*

Comment: Much of the discussion lacks a clear message. From the reading of the Introduction. It sounds like the discussion will focus on contextualizing findings to inform ‘preventable spending’ policies. This connection could be much clearer. Recommendations could also be more punctual and not just a list. We suggest significant re-working of this entire section.

Response: *The manuscript was edited and proofread carefully by a professional through service suggested by BMJ.*

Comment: Conclusion – There is no conclusion

Response: *We have added the Conclusion section.*

VERSION 2 – REVIEW

REVIEWER	Andrew Argent Red Cross War Memorial Childrens Hospital, Paediatric Intensive Care Unit
----------	---

GENERAL COMMENTS

General comments

I am grateful to the authors for their positive responses to the reviewers.

It may be worthwhile highlighting that the authors have really only examined the population of hospital inpatients and the costs related to their care. Potentially over time increasing percentages of healthcare expenditure will be related to outpatient care, and it would be really interesting to address the issue of High-cost healthcare users in the outpatient context.

Specific Comments

Title

The title is substantially improved. I wonder if it would be more accurate if it highlighted the fact that this reviewed only inpatient expenditure (I accept that hospital inpatient services utilize a very substantial proportion of overall healthcare expenditure). Would it be useful to refer to “hospital-care users”?

Introduction

Para 3

The authors state that “Leading causes of burdens of disease (cause of death and disability) ...”. It is interesting to note that diseases can certainly cause death and disability, and it is important to measure the impact of those diseases. However the hospital wards and medical systems may have many patients who have long-term chronic diseases or processes which are not directly (or in the short-term) causes of death or disability (particularly if treated appropriately). However, those patients do in fact create a “burden of care” for the healthcare system. Ironically long-term successful management of chronic conditions will add to the healthcare burden, while not causing death and disability. It may be useful to explore some of that reality in either the introduction or the discussion sections. As healthcare services develop the costs of caring for conditions such as allergy (which may not kill many patients, but can create considerable morbidity and burden of care) are likely to increase.

Para 6

The authors refer to: “Research on the HCUs phenomenon under Thailand’s biggest public health insurance scheme, the UCS, is essential for identifying potential measures that are deemed ‘preventable’ spending.” I am concerned that one should be careful about conflating “preventable” spending with high-cost patients. There are different categories of: spending that would have been avoidable if early intervention had taken place; spending that could have been avoided if specific public health interventions could have been implemented. There are categories of spending that are possibly providing “low return on expenditure” e.g. very few QALY’s or equivalent as return for high expenditure. Those are not “preventable”, but rather categories of expenditure where “return on investment” could be much better – but that is a decision that healthcare services and societies need to consider.

Methods

The methods are clear.

Results

	The results are clearly expressed. I would appreciate more detail on what is included in the category “factors influencing health status and contact with health services (e.g. admissions for investigation)” It is interesting that the category “Injury, or poisoning and certain other consequences of external causes” is one of the major groups for the low cost users, but is also a risk factor for being an high cost user”. This could be reviewed in a little more detail. Does this mean that trauma (which is potentially “preventable”) is a driver of high-cost inpatient care? It is also one of those areas which may have long-term costs beyond healthcare expenditure. Discussion This has substantially improved. It would be interesting to ask the question as to whether there are people in Thailand who are “in need” of high costs interventions and therapies, but who currently do not have access to those therapies? Increasing accessibility of healthcare and therapies potentially could increase this category of patients substantially. Conclusions Appropriate
--	---

REVIEWER	Felicia Marie Knaul University of Miami, Institute for Advanced Study of the Americas
REVIEW RETURNED	14-Jun-2021

GENERAL COMMENTS	Comments from Second Review (Manuscript ID bmjopen-2020-047330): Reviewer 2 Reviewer 2: The authors have conducted a thorough analysis of high-cost users in the context of UHC in Thailand, adding significantly to the literature on this subject in LMICs. After a second review of the article, we have a few comments that should be addressed before resubmission and ultimately publication – we do recommend publication once these have been addressed:  1. The authors have added six references to the manuscript. Four of the new references provide context to the income group classification and growing population of Thailand. We thank the authors for including two references about the Charlson comorbidity index. Even with the new references, we believe that the authors should include some citations from the literature on disparities in health care utilization and UHC, as we well as some on high-cost patients in LMICs.
---

This will make it possible to understand gaps in the literature and also emphasize the value of this study.

2. While there has been improvement, there are still formatting and grammatical errors throughout the document. For example, there are numerous repeated words and inconsistent spacing. There also seems to be a missing reference on line 30.

Comment: Please restate research questions as clear statements

Response: *We have revised the last paragraph of the "Introduction" section.*

Comment #2: The research question is written clearly now.

Comment: While the references that are used are appropriate, many references are lacking throughout the section and the entire paper, beginning with the first couple of sentences.

Response: *The manuscript was edited and proofread carefully by a professional through service suggested by BMJ.*

Comment #2: There are still more references that could be added throughout the paper. Also, the reference to the income group classification of Thailand (#13) should be moved to line 19-20 where the classification is first mentioned.

Comment: We suggest including literature on the disparities in utilization among users within the context of UHC. There are important studies on utilization that need to be referenced as well as integrating literature on high-cost patients. Example: Peltzer, Karl, et al. "Universal health coverage in emerging economies: findings on health care utilization by older adults in China, Ghana, India, Mexico, the Russian Federation, and South Africa." *Global health action* 7.1 (2014): 25314. There is also extensive literature on high-cost patients.

Response: *We thank the reviewer for suggesting the literature and have added accordingly. There are indeed a great deal of literature on health care utilization including high burden diseases or interventions in LMICs. With that being, the literature specifically on high-cost users in LMICs remain limited. We have elaborated this point further in the "Introduction" section. This is another reason we would like to publish our study with the hope that it could be an*

	example and encourage other country (especially LMICs) to fill in this gap in the literature. Comment #2: Thank for including the reference we proposed as an example. We still feel that more literature could be added that could provide more context to health care utilization in the context of UHC and on high-cost users in LMICs. The text added to the introduction only notes that this literature exists.
--	---

VERSION 2 – AUTHOR RESPONSE

Responses to comments by reviewers (Manuscript ID bmjopen-2020-047330.R1)

Reviewer 1 comment : I am grateful to the authors for their positive responses to the reviewers. It may be worthwhile highlighting that the authors have really only examined the population of hospital inpatients and the costs related to their care. Potentially over time increasing percentages of healthcare expenditure will be related to outpatient care, and it would be really interesting to address the issue of High-cost healthcare users in the outpatient context.

Author response : *We agree with your point and believe that this study could be a value to future research to explore the impact of HCU in outpatient care. The focus on hospital inpatients costs has been emphasized in the manuscript. Additionally, we have revised the title of the manuscript to be “Retrospective secondary data analysis to identify high-cost users in inpatient department of hospitals in Thailand, a middle-income country with universal healthcare coverage”.*

Reviewer 1 comment : The title is substantially improved. I wonder if it would be more accurate if it highlighted the fact that this reviewed only inpatient expenditure (I accept that hospital inpatient services utilize a very substantial proportion of overall healthcare expenditure). Would it be useful to refer to “hospital-care users”?

Author response : *The title of the manuscript has been revised to explicitly highlight the inpatient aspect. The title is now “Retrospective secondary data analysis to identify high-cost users in inpatient department of hospitals in Thailand, a middle-income country with universal healthcare coverage”. We would like to keep the term ‘high-cost users’ to be consistent the published literature.*

Reviewer 1 comment : Introduction Paragraph 3, The authors state that “Leading causes of burdens of disease (cause of death and disability) ...”. It is interesting to note that diseases can certainly cause death and disability, and it is important to measure the impact of those diseases. However the hospital wards and medical systems may have many patients who have long-term chronic diseases or processes which are not directly (or in the short-term) causes of death or disability (particularly if treated appropriately). However, those patients do in fact create a “burden of care” for the healthcare system. Ironically long-term successful management of chronic conditions will add to the healthcare burden, while not causing death and disability. It may be useful to explore

some of that reality in either the introduction or the discussion sections. As healthcare services develop the costs of caring for conditions such as allergy (which may not kill many patients, but can create considerable morbidity and burden of care) are likely to increase.

Author response : *Thank you for your suggestion. This point has been addressed in the “Discussion” section (2nd paragraph).*

Reviewer 1 comment : Introduction Paragraph 6, The authors refer to: “Research on the HCUs phenomenon under Thailand’s biggest public health insurance scheme, the UCS, is essential for identifying potential measures that are deemed ‘preventable’ spending.” I am concerned that one should be careful about conflating “preventable” spending with high-cost patients. There are different categories of: spending that would have been avoidable if early intervention had taken place; spending that could have been avoided if specific public health interventions could have been implemented. There are categories of spending that are possibly providing “low return on expenditure” e.g. very few QALY’s or equivalent as return for high expenditure. Those are not “preventable”, but rather categories of expenditure where “return on investment” could be much better – but that is a decision that healthcare services and societies need to consider.

Author response : *We note this good point and have revised the “Introduction” section (6th paragraph) accordingly.*

Reviewer 1 comment : I would appreciate more detail on what is included in the category “factors influencing health status and contact with health services (e.g. admissions for investigation)”

Author response : *Based on the International Statistical Classification of Diseases and Related Health Problems (ICD), this category (Z00-Z99 Factors influencing health status and contact with health services) is provided for occasions when circumstances other than a disease, injury or external cause classifiable to categories A00-Y89 are recorded as “diagnoses” or “problems”. Reference is <https://icd.who.int/browse10/2016/en>.*

Reviewer 1 comment : It is interesting that the category “Injury, or poisoning and certain other consequences of external causes” is one of the major groups for the low cost users, but is also a risk factor for being an high cost user”. This could be reviewed in a little more detail. Does this mean that trauma (which is potentially “preventable”) is a driver of high-cost inpatient care? It is also one of those areas which may have long-term costs beyond healthcare expenditure.

Author response : *We appreciate the reviewer’s point. There are a few points we would like to explain and highlight. First, “Injury, or poisoning and certain other consequences of external causes” or Injury is one of the main diagnoses for the low-cost users. When investigated further, many of the cases had low cost whereas some cases could have high cost. Second, patients with Injury had different demographic characteristics and health service utilization pattern compared with other patients. For example, compared with HCUs with other diagnoses, HCUs with Injury were younger and had fewer number of average visits. After the other explanatory variables being controlled, Injury became one of the predictors for HCU in the regression results as shown in Table 3.*

Third, one interesting point to further explore is that, if not managed well, whether LCUs with Injury can become HCUs later, given Injury is one of the top 5 diagnoses for LCUs and a predictor for HCUs as well. Certain long-term or more severe conditions, e.g. trauma, may also be prevented if patients with Injury are managed well at the early stage. But, we need longer period data to address this question and can be explored in future research. We have elaborated further in the Methods (the fourth paragraph, page 7) and Discussion (the first paragraph, page 18) sections. We agree that this condition (Injury or poisoning and certain other consequences of external causes) may have long-term costs beyond healthcare expenditure; and interesting topic for future research.

Reviewer 1 comment : It would be interesting to ask the question as to whether there are people in Thailand who are “in need” of high costs interventions and therapies, but who currently do not have access to those therapies? Increasing accessibility of healthcare and therapies potentially could increase this category of patients substantially.

Author response : Ideally, we hope that there are no such patients who is in need of high costs interventions and therapies but could not access to them based on Thailand’s Universal Health Coverage since 2002 (1). By law all Thais are eligible for free health care if needed. With that being said, in the real-world, there certainly are potential scenarios where patients may not have access to high cost interventions. For example, some may not have access in a timely manner due to a long waiting list. Some may not be qualified due to certain eligibility criteria, e.g. age, comorbidities. To explore who are truly in need would be an interesting area for future research to study further.

Reference:

(1) Tangcharoensathien, V., Tisayaticom, K., Suphanchaimat, R., Vongmongkol, V., Viriyathorn, S., & Limwattananon, S. (2020). Financial risk protection of Thailand’s universal health coverage: results from series of national household surveys between 1996 and 2015. *International Journal for Equity in Health*, 19(1), 1-12.

Reviewer 2 comment : The authors have conducted a thorough analysis of high-cost users in the context of UHC in Thailand, adding significantly to the literature on this subject in LMICs. After a second review of the article, we have a few comments that should be addressed before resubmission and ultimately publication – we do recommend publication once these have been addressed. The authors have added six references to the manuscript. Four of the new references provide context to the income group classification and growing population of Thailand. We thank the authors for including two references about the Charlson comorbidity index. Even with the new references, we believe that the authors should include some citations from the literature on disparities in health care utilization and UHC, as well as some on high-cost patients in LMICs. This will make it possible to understand gaps in the literature and also emphasize the value of this study.

Author response : *We thank the reviewer for the comment. From our search, we have added relevant literature addressing disparities in health care utilization and UHC as indicated in*

references 19-21. We conducted a search for references informing on high-cost user (HCU) in LMICs. There was a systematic review of HCU study, published in 2018, which identified 55 relevant studies; of which none of them are from LMICs (2). In addition, we have attempted to identify studies published after 2018 from LMICs but could not locate any HCU study in LMICs. We hope our work could add to this gap in the literature.

Reference:

2) Wammes JJ, van der Wees PJ, Tanke MA, Westert GP, Jeurissen PP. Systematic review of high-cost patients' characteristics and healthcare utilisation. *BMJ open*. 2018 Sep 1;8(9):e023113.

Reviewer 2 comment : While there has been improvement, there are still formatting and grammatical errors throughout the document. For example, there are numerous repeated words and inconsistent spacing. There also seems to be a missing reference on line 30.

Author response: *We wanted to ensure that our paper was edited properly. Therefore, we paid to send the manuscript to be edited and proofread carefully by a professional through a service suggested by BMJ. We double-checked the references and proofread again, and hope that this version is acceptable. Thank you.*

Reviewer 2 comment : There are still more references that could be added throughout the paper. Also, the reference to the income group classification of Thailand (#13) should be moved to line 19-20 where the classification is first mentioned.

Author response: *We have added 4 new references in the revised manuscript. The reference number 13 is about a HCU study in Australia. We believe you mean the reference number 22, so that we moved it as suggested.*

Additional revision:

The authors have updated Table 1 (descriptive results on average annual cost) due to a recent discovery of a coding error. The average hospitalization cost per year for both groups (HCUs and LCUs) were slightly lower (please see updated Table 1). The other results remain the same. We apologize for this oversight and thank the editor and reviewers for your understanding.

VERSION 3 – REVIEW

REVIEWER	Andrew Argent Red Cross War Memorial Childrens Hospital, Paediatric Intensive Care Unit
REVIEW RETURNED	30-Jun-2021
GENERAL COMMENTS	I appreciate the response of the authors to comments from the reviewers. I have no additional comments on the content of the paper. There are sections of the manuscript where the flow of the language could be improved. As an example "This study raises an interesting point in setting health priority using disease burden that has been

	relied on morbidity and mortality outcomes in terms of disability-adjusted life-years ..." and "Also, injury in Thailand is often occurred among young population and ..." There are multiple similar issues through the document, and these sorts of language fluency issues are probably best addressed by editorial staff who are primarily English speaking.
--	---